Journal of
open psychology data

# Data from Message Source Effects on Rejection and Costly Punishment of Criticism Across Cultures

DATA PAPER

J. LUKAS THÜRMER [ID]

SEAN M. MCCREA [ID]

*Author affiliations can be found in the back matter of this article

]u[ ubiquity press

## ABSTRACT

We conducted the first systematic intercultural comparison of the Intergroup Sensitivity Effect (ISE). In a 2 × 2 within-participants design, participants from three countries representative of individualism (Canada), collectivism (China) and honour/face (Japan) responded to four critical comments (a) about their own culture or another culture and (b) by a commenter from the same or another culture. Participants responded to each comment on established self-report scales and could punish the commenter, at a cost to themselves. Finally, participants reported potential moderators and demographic information. We report robustness checks and detailed data descriptors to facilitate secondary analyses and follow-up studies.

**CORRESPONDING AUTHOR:**
**J. Lukas Thürmer**

University of Salzburg & Seeburg Castle University, Austria

lukas.thuermer@plus.ac.at

**KEYWORDS:**
intergroup sensitivity effect (group criticism); intercultural differences; individualism; collectivism; honour/face culture

**TO CITE THIS ARTICLE:**

# (1) BACKGROUND

Worldwide, we witness increased societal, hostile divides between (sub-)groups. Psychological research has identified a candidate process propelling such hostile divides, the Intergroup Sensitivity Effect (ISE): Group members are surprisingly open to criticism from fellow ingroup members but reject the same comments from outgroup sources (reviews by Hornsey, 2005; Hornsey & Esposo, 2009; Thürmer & McCrea, 2023a). The ISE was observed in multiple group contexts, including nationality (Hornsey et al., 2002; Thürmer & McCrea, 2018), race/immigration status (Adelman & Verkuyten, 2020; White et al., 2022), sports teams (Braid et al., 2025), and colleges (O'Dwyer et al., 2002; Thürmer et al., 2019).

Two registered reports attest to the robustness of this effect (McCrea et al., 2022; Thürmer & McCrea, 2021) and recent research has thus started investigating the role of the ISE in societal divides, including political tension (Reiman & Killoran, 2023; Thürmer & McCrea, 2024), the acceptance of COVID health measures (McCrea et al., 2025; McCrea et al., 2024; Thürmer & McCrea, 2022; Thürmer & McCrea, 2023b), and climate-related issues (Thürmer, Bamberg, et al., 2024; Thürmer et al., 2022). Regarding potential explanations of this effect, existing research points to violation of conversational norms (McCrea et al., 2022; Sutton et al., 2006), defending one's group (Hornsey et al., 2004; Thürmer et al., 2019), and attribution of intent (Hornsey & Imani, 2004; Thürmer, Chen, et al., 2024).

All of these approaches would suggest that cultural differences may exist regarding the ISE. Specifically, the key cultural mindsets of collectivism, individualism, and honor (Oyserman, 2017) have been shown to moderate responses to reputational threats and maintaining one's social identity (e.g., Eriksson et al., 2021; Uskul et al., 2023). To test these predictions and fill this outstanding research gap, we conducted a systematic cultural comparison of the ISE.

We had the opportunity to conduct a large-scale registered intercultural experiment ($N$ = 2207). Our pre-registration protocol was reviewed anonymously and, once accepted, funded by the Leibniz foundation. We selected three previously understudied countries that are representative of the key cultural dimensions of individualism (Canada), collectivism (China), and honor (Japan). Participants across cultures indeed rejected outside criticism of their own group more than identical comments from within the group. However, a bystander effect (i.e., when another group was criticized) only emerged in Canada and Japan, but not in China. Apparently, the psychological processes differ between these countries. We identified previously unknown cultural moderators, making a key contribution to understanding societal rifts world-wide.

In the current contribution, we describe the data in detail to facilitate their continued use. To this end, we describe the data collection procedure, the collected measures, and the robustness checks conducted. We hope that this paves the way for secondary analyses and for future research.

## REGISTERED HYPOTHESES AND EXPLORATORY ANALYSES

We designed the study to test the following registered hypotheses and research questions (Thürmer et al., 2023):

H1.) Consistent with the normative account, any type of intergroup criticism will result in more negative responses to the message (i.e., motive and threat) and the commenter (commenter ratings and reported anger) compared to intragroup criticism. Thus, we expect a main effect of source of the comment on these self-report and behavioral measures (i.e., lottery allocation).

H2.) We provide two competing hypotheses regarding culture, based on the available accounts of the ISE:

   **a)** According to a norm account, we expect a Source × Culture interaction, such that people from Japan [honor] show a greater classic and bystander ISE (i.e., Source effect) than participants from China [collectivist] and Canada [individualist].

   **b)** According to a social identity account, we expect a Source × Target × Culture interaction, such that people from China [collectivist] show a greater classic ISE than participants from Canada [individualist] and Japan [honor]; this effect should not emerge for the bystander ISE.

   E1: We will compare classic and bystander ISE between cultures

   E2: We explore a potential moderation of the ISE by perceived appropriateness of criticizing outgroups (i.e., norms) using Johnson-Neyman analyses.

   E3: We explore a potential moderation of the ISE by national identification using Johnson-Neyman analyses.

   E4: We will use equivalence tests (Lakens et al., 2018) for non-significant predicted effects to establish whether a practically relevant effect could exist.

# (2) METHODS

## 2.1 STUDY DESIGN

The experiment followed a 2 Comment Target (participants' own cultural group vs. another cultural group) × 2 Comment Source (same cultural group as

criticized [within-group criticism] vs. other cultural group as criticized [between-group criticism]) within-participants design. Accordingly, participants read four comments that each were (a) about their own culture versus another culture and (b) by a commenter from the same versus another culture. As a quasi-experimental between participants factor (Culture), we recruited participants from Canada, China, and Japan.

The order of the comments and conditions was fully randomized by the survey software formR (Arslan et al., 2020). To this end, participants were assigned to one of four counterbalancing orders for the comment content (variable rand in the dataset), for the condition (variable cond in the dataset) and which other country was assigned to be the outgroup target and outgroup source (variable outrand in the dataset). For the within-participants randomizations of factors with more than two stages (i.e., rand and cond), orders were determined according to a Latin-square design (Tables 1 and 2). In sum, the data collection procedure was fully automated and the study team did not interact with participants during the study.

### Translation Procedures
Native speakers in Japanese and Taiwanese (which is derived from standard Mandarin Chinese) who were all highly proficient in English provided the translation from the English study materials. The translators met with the first author to discuss the aims of the study and the characteristics of the study materials. Ambiguities were discussed whenever they emerged. After completing the first translation. The first author used machine-aided back translation to identify deviations from the original study materials. Whenever such deviations emerged, they were discussed with the translators. All deviations were resolved in a satisfactory manner.

### 2.2 TIME OF DATA COLLECTION
We collected data between 03/17/2023 and 03/29/2023 using the professional panel provider Respondi (respondi.com) that was hired directly by the funder Leibniz foundation.

### 2.3 LOCATION OF DATA COLLECTION
Data was collected in Canada, China, and Japan, through the panel provider Respondi.

### 2.4 SAMPLING, SAMPLE AND DATA COLLECTION
#### Power
Previous research has observed large ISEs on self-report measures (typically between $d = .630$ and $d = 1.005$) and moderate behavioral ISEs (e.g., $d = .269$ in Thürmer & McCrea, 2018). We used G*Power (Faul et al., 2009) to conduct an a priori power analysis for our $2 \times 2 \times 3$ mixed-measures ANOVA, assuming a small effect of $f = 0.046$ and setting $1-\beta = .95$, with $\alpha = .05$. This power analysis indicated a required sample size of $N = 1559$. We aimed to recruit $N = 1800$ participants, to account for possible dropouts (see below).

#### Sampling
We applied to the Lab Track at the Leibniz Institute for Psychology (PsychLab online) to fund the data collection. We were unable to stratify our sample according to a full demographic survey (e.g., age, gender, education etc.) because this would have led to extremely small subgroup sizes; we thus relied on stratification by binary gender (Table 3). However, the sample allows for exploring moderating effects of gender identity, given sufficient subsample sizes for the question. To this end, we followed the suggested UNECE guidelines (https://unece.org/fileadmin/DAM/stats/publications/2019/Issue_5_Gender_Identity.pdf), and included three gender options (i.e., "male", "female", "other/non-binary") in the study questionnaire.

| ROUND | RAND == 1 | RAND == 2 | RAND == 3 | RAND == 4 |
|---|---|---|---|---|
| Round 1 | Comment 1 | Comment 2 | Comment 3 | Comment 4 |
| Round 2 | Comment 2 | Comment 1 | Comment 4 | Comment 3 |
| Round 3 | Comment 3 | Comment 4 | Comment 1 | Comment 2 |
| Round 4 | Comment 4 | Comment 3 | Comment 2 | Comment 1 |

**Table 1** Counterbalancing conditions (variable rand) for the order of the comment content.

| ROUND | COND == 1 | COND == 2 | COND == 3 | COND == 4 |
|---|---|---|---|---|
| Round 1 | Ingroup target; Within-group source | Ingroup target; Intergroup source | Outgroup target; Within-group source | Outgroup target; Intergroup source |
| Round 2 | Ingroup target; Intergroup source | Ingroup target; Within-group source | Outgroup target; Intergroup source | Outgroup target; Within-group source |
| Round 3 | Outgroup target; Within-group source | Outgroup target; Intergroup source | Ingroup target; Within-group source | Ingroup target; Intergroup source |
| Round 4 | Outgroup target; Intergroup source | Outgroup target; Within-group source | Ingroup target; Intergroup source | Ingroup target; Within-group source |

**Table 2** Counterbalancing conditions (variable cond) for the order of the message source and target conditions.

| GEOGRAPHICAL LOCATION (COUNTRY) | CULTURAL MINDSET | FEMALE | MALE | OTHER |
|---|---|---|---|---|
| Canada | Individualistic | 300 | 300 | – |
| China | Collectivism | 300 | 300 | – |
| Japan | Honor/Face | 300 | 300 | – |

**Table 3** Planned minimum sample size by country and gender.

| GEOGRAPHICAL LOCATION (COUNTRY) | CULTURAL MINDSET | FEMALE | MALE | OTHER |
|---|---|---|---|---|
| Canada | Individualistic | 340 | 325 | 1 |
| China | Collectivism | 314 | 472 | 0 |
| Japan | Honor/Face | 307 | 448 | 0 |

**Table 4** Actual sample sizes by Country and gender (after exclusions).

| GEOGRAPHICAL LOCATION (COUNTRY) | CULTURAL MINDSET | MEAN | SD | MEDIAN | MIN; MAX |
|---|---|---|---|---|---|
| Canada | Individualistic | 45.71 | 12.24 | 45 | 17; 69 |
| China | Collectivism | 45.69 | 11.36 | 45 | 18; 75 |
| Japan | Honor/Face | 49.12 | 12.08 | 50 | 18; 69 |

**Table 5** Age by Country (after exclusions).

Participants responded to the panel provider and were paid a fixed amount according to the panel provider guidelines. In addition, participants could win a lottery drawing for a bonus in their respective currency (C$70; JP¥7000, CN¥350). Participants were informed of that payment ahead of their participation.

### Exclusions

The survey software formR was programmed to check manipulation check/reading check responses after the first two comments (Rounds 1 and 2) and the last two comments (Rounds 3 and 4). Participants responded to two manipulation check/reading check items (see above). Participants who responded to more than one item in each block incorrectly were screened out and could not complete the survey. Their data was discarded.

Moreover, as pre-registered, data from participants who did not finish the survey for other reasons, responded to one page of the questionnaire consistently with the same scale value (e.g., all 1; *straight liners* as identified with the *careless* package (Yentes & Wilhelm, 2023), or participated repeatedly (as indicated by their IP address) were excluded from analysis. No further exclusion criteria beyond those registered were applied.

### Final Sample

A total of 2462 participants completed our study (our dataset contains 2930 incomplete study entries, including those who failed attention checks and could not continue [see below]). As preregistered, we excluded 194 participants for giving the same response on all items on at least one questionnaire page ("straight liners") and 61 for participating repeatedly (as indicated by their IP address), leaving $N = 2207$ for analysis. Due to a programming error, participants could check the consent box and then uncheck it or proceed without checking it. This was the case for 135 participants who completed the entire study (i.e., their informed consent variable was recorded as 0 instead of 1). The local ethics committee determined that participants' completion of the study provides sufficient evidence of their consent to participate; accordingly, we retained their data.

The final sample reached the minimum planned sample size in all cells of the design (Table 4). As expected, only few participants identified the gender category "other", precluding any systematic analysis on this dimension.

Average age of participants was quite similar between the studied countries, with Japanese participants being somewhat older (Table 5). This could reflect the older average age of the Japanese population, in comparison with the Canadian and Chinese population. In line with this view, the sample means from Canada ($M = 45.71$) significantly differed from the population mean ($M = 41.6$), $t(656) = 8.67$, $p < .001$, as did the Chinese sample mean ($M = 45.69$; population $M = 39.6$), $t(785) = 15.03$, $p < .001$; but no difference between the Japanese sample mean ($M = 49.12$) and the population mean ($M = 49.4$) was observed, $t(755) = –0.64$, $p = .520$. Interestingly, the age means of our sample were more similar between cultures than the respective populations. In line with this assessment, observed age ranges were comparable between subsamples. In all, the drawn samples were comparable between countries.

### 2.5 MATERIALS/SURVEY INSTRUMENTS

Participants responded to the panel provider's study advertisements by clicking on a web-link to a web-based survey in formR (Arslan et al., 2020). Participants first responded to questions regarding their main residence (with the three flags and "other" displayed as options) and gender ("Please indicate your gender:" with the options "male", "female", and "other"). Participants could only continue if they indicated the respective nationality of the target sample. Binary sex assigned at birth (male/female) was used for stratification purposes. Participants indicating "other" could continue and were not counted towards the quota.

The initial screening questions were displayed in all three languages (English, Chinese, and Japanese); subsequent materials were only provided in participants' respective language. Participants were told that we need their help in rating comments by previous participants

in a study on people's thoughts about international teamwork in times of globalization and learn about their personal opinion. This procedure has been used in a host of previous ISE research (e.g, Thürmer & McCrea, 2018, 2021, 2022), and the materials of this study were directly adapted from McCrea et al. (2022), a previous registered report conducted with US participants (McCrea et al., 2022). This way, we ensured a stringent intercultural comparison, such that any deviation in results would likely be due to cultural differences and not different study materials. Participants read critical comments describing people from the respective target country as difficult to work with. We constructed four such comments that were all very similar in length (62–64 words; see Stimulus Material below). The order of the comments and the order of the conditions was fully counterbalanced (randomized; see Tables 1 and 2 above).

The comment either criticized participants' own national group (ingroup target) or another national group (outgroup target). To manipulate the message source, the comment was attributed to a former participant from the country targeted in the comment (within-group source condition) or another country (intergroup source condition). After reading each comment, participants indicated the nationality of the commenter and the target group of the comment, as a manipulation/reading checks, and responded to the comment on established scales (Message Motive, Message Threat, Commenter Evaluation, and Anger; see Scale Measures: Dependent Variables below). After these ratings, participants had the opportunity to punish the commenter by denying lottery tickets in a drawing for a bonus in the respective currency (C$70; JP¥7000, CN¥350; see Behavioral Measure: Dependent Variable below).

After the last comment, participants indicated their social identification with their own country as well as their connection to the other two countries involved in the study. Participants' own cultural orientation was assessed with scales designed to measure individualism/collectivism with the subscales horizontal individualism (HI), vertical individualism (VI), horizontal collectivism (HC), and vertical collectivism (VC), adapted from Triandis and Gelfand (1998) and honor, adapted from Novin and Oyserman (2016). Participants also indicated how appropriate they found different forms of group criticism and defense, adapted from McCrea et al. (2022). Scale instructions and items are discussed below (see Scale Measures: Potential Moderators).

Participants provided demographic information ("Some final questions about you:"), including their age ("Please indicate your age:"), their nationality ("What is your nationality?" with the options "Japan", "China, "Canada", and "other (please specify):"), how long they have resided in their country ("How long have you been living in the indicated country?"), and their association with their country ("How much do you associate yourself with this country?" on a 7-point scale from 1: Not at all to

7:Very much). Finally, participants were asked about the study goals ("What do you think the study was about?") and were fully debriefed about the employed deception regarding the comment source (i.e., that we in fact wrote the comments) and given the opportunity to withdraw their consent, which none of the participants opted to do.

## Stimulus Materials: Comments

To ensure the comparability with existing ISE research, we directly adapted the critical comments from a published registered report conducted with US-American participants (McCrea et al., 2022). As in the original study, comments were identical across conditions, with the only exception that within-group criticism used we-phrasing and intergroup criticism used they-phrasing. As highlighted below, the source nationality was highlighted in the introductory sentence, and the respective national flag was displayed above the comment. Intergroup comment source phrasing is provided in parantheses.

### Comment 1
[source national flag]
The following comment was written by a former participant from [source nationality]
"In my experience, people from [target nationality] are unbelievable stubborn and inflexible. We (They) are really hard to talk to. Our (Their) own opinion counts more than anything, no matter how stupid or obviously wrong it is. I recommend to never even start a smart discussion with a person from [target nationality]. To win an argument, we (they) will stick with our (their) own stereotypes and wrong information."

### Comment 2
[source national flag]
The following comment was written by a former participant from [source nationality]
"To be honest, people from [target nationality] are actually dim-witted. We (They) make believe that education and knowledge is important, but most of us (them) are rather unintelligent. And on top of that we (they) are most of the time easily tricked. It's clear that people from [target nationality] are rather simple minded, and it makes me sometimes so ashamed to see how dumb we (they) are."

### Comment 3
[source national flag]
The following comment was written by a former participant from [source nationality]
"People from [target nationality] cannot be trusted. We (They) are just materialistic and arrogant and will do anything to fool you. We (They) really have no honour whatsoever and we (they) will not miss an opportunity to cheat. I know for a fact that people from [target nationality] also really like to cheat in sports and in education, that is why we (they) achieve such good results."

*Comment 4*
[source national flag]
The following comment was written by a former participant from [source nationality]

"People from [target nationality] are so rude and careless, and we (they) really only look out for ourselves (themselves). It is so embarrassing that we (they) are so obnoxious and standoffish, especially how we (they) think we (they) are superior, when most of us (them) are just really lazy. People from [target nationality] will find any means to avoid work and are happy to let others work for us (them)."

## Scale Measures: Dependent Variables

Participants rated the message motive (3 items: "To what extent do you think" and endings about the respective targeted group: "the comments were constructive," "the person who wrote these comments cares about [Country]," "the comments were made in [Country's] best interest"), assessed the message threat (8 items; "To what extent do you think this comment is" and the selection of: "threatening," "disappointing," "irritating," "offensive," "insulting," "hypocritical," "judgmental," "arrogant"), evaluated the commenter (7 items; "To what extent do you think the commenter is" and these endings: "intelligent," "trustworthy," "friendly," "open-minded," "likable," "respectable," "interesting"), and indicated how much anger they felt towards the commenter ("Rate how you feel about the commenter:"; "How angry are you with the commenter?"; "How furious are you with the commenter?"; "How irritated are you with the commenter?"). The scales were directly adapted from previous ISE research (Hornsey & Imani, 2004; McCrea et al., 2022; Thürmer & McCrea, 2021) and all employed a 7-point Likert scale (1 = *not at all* to 7 = *very much*).

## Behavioral Measure: Dependent Variable

Finally, participants had the opportunity to deny the commenter lottery tickets in a drawing for a bonus (equivalent of €50 in local currency; C$70; JP¥7000, CN¥350) on a 7-point scale (0 lottery tickets to 12 lottery tickets; increments of two). To make punishment costly, participants were informed that they would receive 5 lottery tickets as a bonus plus half of the commenter's lottery ticket bonus for their help. Therefore, participants were able to earn between 5 and 11 lottery tickets per comment. After the study was complete, the number of lottery tickets per participant was assessed, one winner ticket was randomly drawn per country, and the respective bonus was paid via the sample provider. The item was directly adapted from McCrea et al. (2022) and read:

*"We offered a bonus of up to 12 lottery tickets to previous participants who provided high quality responses in this study. The lottery tickets are placed in a drawing for the chance to win a C$70*

*bonus. We would like you to judge the quality of the comment above and indicate how much of a bonus we should give to this person.*
*Important: In order to ensure that this process is fair, we will (a) give you a fixed bonus of 5 lottery tickets for your help in the bonus decision and (b) half of the lottery tickets that the other participant receives. Please give your honest assessment of the quality of the comment."*

The provided responses options were: "Excellent comment that deserves the full bonus of 12 tickets", "Good comment that deserves a large bonus of 8 tickets", "Reasonable comment that deserves a bonus of 6 tickets", "Deficient comment that still deserves a small bonus of 4 tickets", and "Unacceptable comment that deserves 0 tickets."

## Scale Measures: Potential Moderators

At the end of the experiment, participants were asked to complete a measure of ingroup identification, responding on a seven-point Likert-scale (1 = *strongly disagree* to 7 = *strongly agree*) on the items: ("Overall, being [nationality] has very little to do with how I feel about myself" [reverse coded]; "Being [nationality] is an important reflection of who I am"; "Being [nationality] is unimportant to my sense of what kind of a person I am" [reverse coded]; "In general, being [Country] is an important part of my self-image"; adapted from Leach et al., 2008). Also using 7-point scales, participants will indicate their connection to the other two studied countries ("I care deeply about [Country]."; "[Country] is important to me."; "I have a strong connection to [Country]."; "Do you like [Country]?").

Afterwards, participants were asked to respond regarding their connection to the outgroups used in this research (e.g., for participant from Japan, the outgroups were China/Canada) on two three-item scales ("I care deeply about China/Canada/Japan; China/Canada/Japan is important to me; I have a strong connection to China/Canada/Japan") on a seven-point scale (1 = *strongly disagree* to 7 = *strongly agree*).

This was followed by an assessment of their estimated norms towards appropriateness of intergroup criticism and ingroup defensiveness (adapted from McCrea et al., 2022; Sutton et al., 2006). Participants were asked to indicate on a seven-point scale (1 = *not at all*, 7 = *very much*) to what extent it is appropriate for people to "criticize groups to which they themselves belong?"; "criticize groups to which they do not belong?"; and "defend their own group from criticism by outsiders?".

## 2.6 QUALITY CONTROL
### Counterbalancing Checks

To evaluate our counterbalancing, we tested for potential effects of message content and ordering. We preregistered that, should any unexpected effects arise, we would standardize the dependent measures

according to the respective counterbalancing factor using z-transformations. Some order effects emerged, and we accordingly z-scored all response scales by round (i.e., subtracted the respective Round average and divided by the respective Round standard deviation).

## Data Pre-Processing

To prepare the dataset for analyses, results from the different language versions of the questionnaires were merged. Average scale scores were computed for all measures with multiple items and the reliability of the scales using Cronbach's α (see Reliability Analyses). All scales were adapted from previous research and have been observed to be reliable; we thus expected satisfactory scale reliabilities. We pre-registered that we would exclude scale items if, contrary to expectations, the reliability of a scale should be α < .70 and if item exclusions would substantially increase reliability. As outlined below, this was not the case.

## Reliability Analyses

We computed single indexes for message motive (α = .89–.92), message threat (α = .92–.93), commenter evaluations (α = .96–.97), anger (α = .96–.97), identification with own country (α = .64; excluding items did not improve reliability; hence, all items were retained), importance of the other countries (α = .89–.91), the individualism-collectivism subscales (horizontal individualism α = .86, vertical individualism α = .80, horizontal collectivism α = .85, and vertical collectivism α = .86), and honor (α = .92).

## Robustness Checks

In the Chinese sample, one dependent measure was not displayed correctly in 3 out of 4 trials. In these trials, the bonus was promised as "cent" instead of "lottery tickets". Our consistency checks indicate that this error did not affect our results (see Thürmer et al., 2025 and Figure 1). Corroborating this view, we observed consistent effects on the bonus measure and the self-

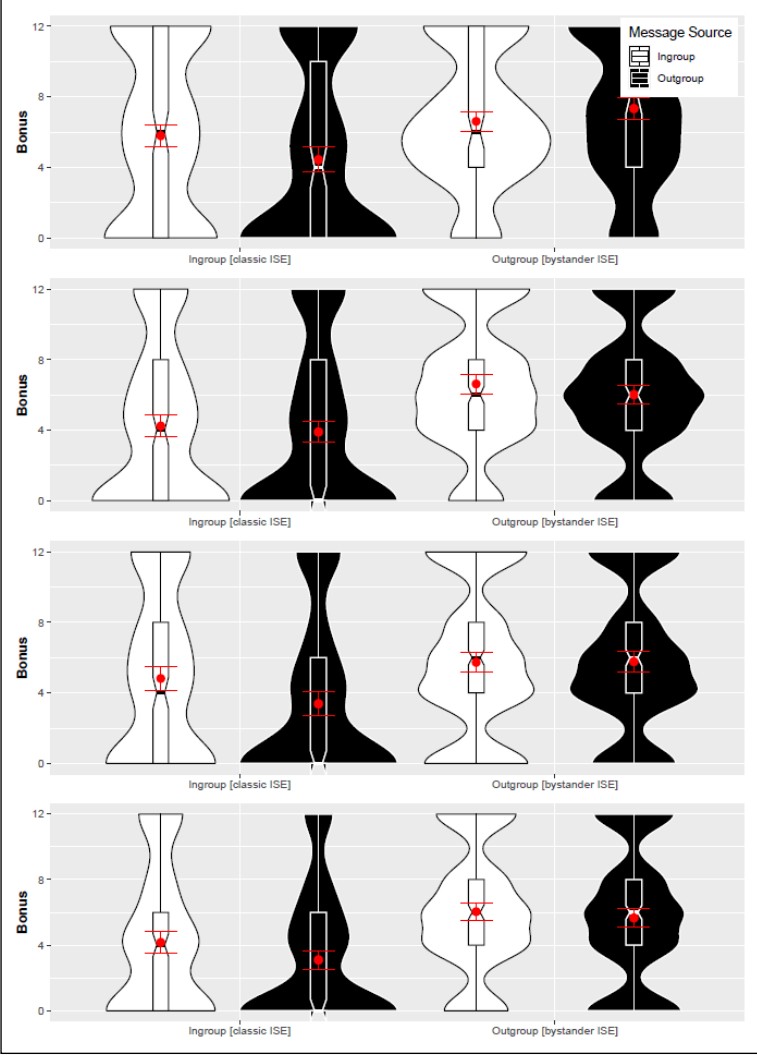

**Figure 1** Boxplots with Violin-Plots: Message source (ingroup vs. outgroup) and Message target (ingroup vs. outgroup) by Round in the Chinese sample. *Note*: Red dots with lines indicate means with 95% CIs. Bonus: Number of lottery tickets awarded to commenter. Due to a programming error, Chinese participants saw "ct" instead of "lottery tickets" when making their bonus decision in Rounds 1 to 3 (red frame); Round 4 (green frame) was displayed correctly as "lottery tickets". Bonus items were displayed correctly in all trials in Canada and Japan.

report dependent measures that were all displayed correctly.

## 2.7 DATA ANONYMISATION AND ETHICAL ISSUES

The University of Salzburg internal review board approved this study (GZ 10/2020).

To anonymize data for sharing, panel identifiers, IP addresses and open-ended responses were removed from the dataset.

## 2.8 EXISTING USE OF DATA

The primary hypothesis tests (see above) are reported in one paper to date (Thürmer et al., 2025).

## (3) DATASET DESCRIPTION AND ACCESS

### 3.1 REPOSITORY LOCATION

The project is publicly available at https://osf.io/djkz8 with the DOI https://www.doi.org/10.17605/OSF.IO/DJKZ8.

### 3.2 OBJECT/FILE NAME

ISE_Culture_Analysis.csv

### 3.3 DATA TYPE

Primary data.

### 3.4 FORMAT NAMES AND VERSIONS

Data are provided in .csv format and can be accessed with most tabulation software packages (e.g., Excel, OpenOffice) as well as statistical software (e.g., R, SPSS). Analyses are provided as R scripts and results as pdf print-outs. The survey materials and a codebook are provided as .xlsx sheets which can be imported in formR but are also readable with most tabulation software packages (e.g., Excel, OpenOffice).

### 3.5 LANGUAGE

The data are in American English. The materials are in American English, Japanese and Standard Mandarin Chinese.

### 3.6 LICENSE

CC-By Attribution 4.0 International.

### 3.7 LIMITS TO SHARING

Code for the survey setup in formR and the analyses of the original publication are available at https://osf.io/djkz8 (i.e., no limits to sharing).

### 3.8 PUBLICATION DATE

The dataset was uploaded in OSF on June 17, 2024, made public on March 25, 2025, and updated on July 28, 2025, based on comments from reviewers. The project DOI was created on September 28, 2025.

### 3.9 FAIR DATA/CODEBOOK

We provide all survey materials in the easily accessible .xlsx format. Specifically, the variable names of the collected items are stored in the "name" column, the item wording in the "label" column, and the responses option(s) in the "choice" column(s). The file ISE_Culture_Codebook.xlsx further summarizes the variables in the data file. Our analysis scripts refer to these variable names, and our compound measures follow the same taxonomy. We further use file formats widely accessible across all major computing platforms and a broadly used repository (OSF). For our dataset preparation and analyses, we provide detailed R-scripts as well as pdf documentation of our results. Accordingly, our data are findable, accessible, interoperable and our results reproducible.

## (4) REUSE POTENTIAL

Our data provides ample opportunities for follow-up analyses, for new analyses in combination with other existing data and for new experiments. For follow-up analyses, there are a number of potential comparisons in our design that we did not test because they were not relevant to our main question. For instance, analyzing age and gender influences on the Intergroup Sensitivity Effect (ISE) in the existing data can provide valuable insights, especially if paired with other existing datasets of ISE studies. Age and gender have not been considered as a systematic moderator of the ISE but may well have an impact. Younger participants might be more open to criticism compared to older participants, potentially due to generational differences in social norms and exposure to diverse viewpoints. Gender differences could also play a role, with men and women potentially reacting differently to intergroup criticism based on socialization patterns and gender norms. Understanding these demographic influences can help tailor interventions to specific age and gender groups, making them more effective. Further exploring the interplay of cultural dimensions and participant location is promising. For instance, further cultural analyses could highlight how the interplay of participant location and reported cultural preferences shapes the ISE. By analyzing these cultural orientations, researchers can identify specific cultural factors that moderate the ISE and develop culturally sensitive strategies to address intergroup sensitivity.

A further potential lies in correlational analyses, which we have not conducted. Possible open questions, for instance, are the interaction of culture and gender on the evaluation of criticism (independent of the source) or the

relation of culture and identification. Our bonus payment measure could also be used as a general behavioral measure of willingness to pay in such correlational analyses. Finally, our measures of identification, culture, and demographics could be used to test the interplay of these variables. Our sample is sufficiently large to conduct elaborate analyses such as exploratory structural equation modelling. In this regard, our dataset could also be combined with country-level data on other variables of interest. Such variables could include socio-economic factors, country-level cultural differences or political specifics. These factors could then be used as to estimate country-level intergroup sensitivity or in relation to the assessed self-reported characteristics.

Follow-up studies can draw on our methodology and materials. We provide extensively tested materials in three widely-spoken languages. Building on these materials, future research could study the ISE in further cultures by using a similar translation procedure. Moreover, regional differences within the established languages could also be of interest. Our basic research moreover provides the foundation for systematic applied research. Recent ISE research has turned to key societal challenges, including inter-racial conflict (Ireland et al., 2024; White et al., 2022), political division (Reiman & Killoran, 2023; Thürmer & McCrea, 2024), vaccine hesitancy (McCrea et al., 2025; McCrea et al., 2024; Thürmer & McCrea, 2022) and resistance to calls for climate action (Thürmer et al., 2025; Thürmer, Bamberg, et al., 2024; Thürmer et al., 2022). Combining these approaches, the reported data and materials could be used to establish interventions to reduce societal polarization or acceptance of climate action across cultures, thereby addressing world-wide issues. We hope that this data paper will encourage others to us our data in this way and stimulate a host of new research on the rejection of intergroup criticism.

## ACKNOWLEDGEMENTS

We thank Stefanie Müller (ZPID) for her support in the publishing process.

## FUNDING STATEMENT

Data collection of the present study was funded by PsychLab, a service of the Leibniz Institute for Psychology (ZPID). This research was funded in part by the Austrian Science Fund (FWF)[https://doi.org/10.55776/P37261].

## COMPETING INTERESTS

The authors have no competing interests to declare.

## AUTHOR CONTRIBUTIONS

Manuela Wagner, University of Salzburg: Preparation pre-registration and funding application.

Jakob Reichert, University of Salzburg: Preparation of study materials, data collection and analyses.

Kyoko Shinozaki, University of Salzburg: guidance regarding cultural matters.

Wei Wei Chang-Albert, University of Salzburg: Translation of study materials (Mandarin Chinese).

Hikari Beck, University of Salzburg: Translation of study materials (Japanese).

Sean McCrea, University of Wyoming: Concept of study, study materials, preparation of manuscript, writing (editing, revisions).

J. Lukas Thürmer, University of Salzburg and Seeburg Castle University: Concept of study, preparation pre-registration, funding application, preparation of study materials, supervision of translation of study materials, data collection, analyses, writing (initial draft).

## AUTHOR AFFILIATIONS

**J. Lukas Thürmer** orcid.org/0000-0002-5315-2847
University of Salzburg and Seeburg Castle University, Austria
**Sean M. McCrea** orcid.org/0000-0003-4241-5530
University of Wyoming, United States

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

## PEER REVIEW COMMENTS

*Journal of Open Psychology Data* has blind peer review, which is unblinded upon article acceptance. The editorial history of this article can be downloaded here:

- **PR File 1.** Peer Review History. DOI: https://doi.org/10.5334/jopd.133.pr1

**TO CITE THIS ARTICLE:**

**Submitted:** 20 February 2025    **Accepted:** 07 October 2025    **Published:** 15 December 2025

