## [Peer Review History. · Journal of Open Psychology Data]

Reviewer 1:

Recommendation: Revisions Required

Comments to the author(s)

Criteria in the JOPD reviewer guidelines are generally fulfilled. The data paper describes the study design, data collection process and analysis method in rigorous detail. Explicit description of the deposited data is somewhat sparse, but sufficient for interested third parties to be able to access and use it. The data are deposited in a suitable repository, under a suitable license, and in a proprietary but openly accessible format (.xlsx). Column names are sensible and the data format can be understood by consulting the study materials, but a condensed document containing very brief descriptions of each column's content would be helpful. Data processing scripts are valid R and run without error in a blank environment (with one exception, see below). Deposited data carry no identification risk as far as the reviewer can assess.

== Major comments / issues:

Several data files contain rows with consent = 0 or consent = NA. It is not explicitly stated in the study materials or the data paper what these values mean. If it means subjects who did not give consent are included, these should be removed from all data files. If it means something else, the meaning should be documented either in the study materials or data paper. The affected data files are "ISE_Culture_CAN_A.xlsx", "ISE_Culture_CN_A.xlsx", and ""ISE_Culture_JP_A.xlsx".

== Minor comments / issues:

No condensed codebook is provided beyond the "study materials" .xlsx files. These files can generally serve as a codebook to understand the dataset, but some data columns, e.g. of type "calculated", lack a human-readable description in the study materials files. While their variable names are to some extent self-explanatory, more explicit description might help understand the data faster.

In the data paper, participation incentives are not discussed in section 2.4 "sampling, sample and data collection". It is mentioned later, in section 2.5 "Behavioral Measure: Dependent Variable", that a bonus lottery was used as part of the experimental design. This

should already be briefly mentioned in the data collection paragraph, and it should be made clear which (if any) conditional or unconditional participation incentives were offered in addition to the lottery. It should be specified whether and to what extent participants were informed about the lottery ahead of participation.

In the repository, .xlsx files for study materials and data have the same names and therefore cannot be opened simultaneously by current versions of MS Excel. This should be addressed e.g. by adding a discriminator such as “_data”, “_materials” to file names.

The data preparation script “ISE_Intercultural_DatasetPrep_StraightLiner_final.R” stops with an error at line 130 because the code section for identifying IP address duplicates is not applicable to the shared data. In the interest of usability, the code section applicable to the shared data (line 126, currently commented out) should be uncommented, and all lines not applicable to the shared data should be commented out. The description in line 125 should be updated accordingly.

Reviewer 2:

Recommendation: Revisions Required

Comments to the author(s)

In the present data publication the authors describe a very interesting cross-cultural data corpus aimed at the systematic intercultural comparison of the intergroup sensitivity effect. I really enjoyed reading the manuscript. However, I think there need to be addressed some points before the data publication can develop its full potential and the data is available to the community for reuse in accordance with the FAIR criteria. In the following, I will include a detailed description of the points that I noticed during the reception of the manuscript. I will organize them according to the journal's review criteria.

The methods section of the paper must provide sufficient detail that a reader can understand how the dataset was created, and would within reason be able to recreate it.

*The methods section is written in a very sophisticated and comprehensible way so that it is easy to follow. In addition, the provision of concrete materials and item texts ensures that other researchers should be able to easily recreate the data set. However, there are also some minor issues that I would like to point out. First, I was wondering why there are no citations in the background section describing, for example, the different approaches to ISE or defining ISE? Second, I would recommend carefully checking the paper again for typos (e.g. the “a” in ‘Canada’ is missing in section 2.3 or ‘Leibniz’ vs. “Leibnitz”, etc.). Finally, I wondered if it wouldn't be better to rename the second paragraph in the background section to “Registered hypotheses and exploratory analyses”.

The dataset must be correctly described.

*Altogether, I have to admit that I found the dataset to be described quite poorly compared to the provided methods description. First of all the data publication seems to refer to the merged dataset (merged across countries). However, the deposited data are split by country and something else (A, B, C) that I could not align with the described study design. It would be very helpful to provide some information on which data is in these different files, for instance, via a readme file. Moreover, it was my impression that it could be very difficult for other researchers to analyze the data in a meaningful way, as the identifiers have been removed and therefore the data can no longer be analyzed at the participant level. Perhaps the authors should consider replacing the panel identifier, which is a pseudonym, with an arbitrary code. Then data would still be anonymous, but could also be analyzed on a participant level. Moreover, I had the impression that the data sets used in the analysis scripts differed from those provided via the OSF. Regarding I would recommend to provide a little bit more precise information on the deleted open-ended response variables as well as on other anonymization techniques that have been applied (e.g. grouping or replacing information by more general information). Finally, the authors should be aware that their data should be described with rich metadata in order to comply to the FAIR principles.

The reuse section must provide concrete and useful suggestions for reuse of the data.

*The authors describe a variety of valid reuse scenarios. Therefore, I would see this criterion to be fulfilled.

The repository the data is deposited in must be suitable for this subject and have a sustainability model (see our list of recommended repositories).

*It is planned to publish the data via OSF upon acceptance of the paper currently being under review. So this criterion is also fulfilled as soon as the data has been published.

The data must be deposited under an open license that permits unrestricted access (e.g. CC0, CC-BY).

*Currently, data are only accessible via a reviewer link. However, data have already been assigned an open license CC-BY 4.0.

The deposited data must include a version that is in an open, non-proprietary format.

*Data are provided as .xlsx-files, which is a proprietary format. So I would recommend to provide data also as csv-files. Moreover, it would be better to provide all data as single files and not as one zipped file. This is better in terms of findability and also accessibility.

The deposited data must have been labelled in such a way that a 3rd party can make sense of it (e.g. sensible column headers, descriptions in a readme text file).

*Currently, there is no comprehensive codebook for each dataset. Even though the authors provide a codebook for each of the nine deposited datasets, it is quite hard to get the meaning of the variables from the provided descriptions. I would recommend the authors to provide for each variable the following information: variable name, variable label (this should be descriptive), item text/instructions, value labels, missing value labels. Ideally, in the description, care is also taken to ensure that a simple link between the variables and the implemented experimental design is possible.

The deposited data must be actionable – i.e. if a specific script or software is needed to interpret it, this should also be archived and accessible.

*Data should be actionable.

Studies involving human subjects should adhere to local ethical standards at the host institution and follow American Psychological Association's (APA) Ethical Principles of Psychologists and Code of Conduct (<http://www.apa.org/ethics/code/index.aspx>). Participant data should be sufficiently anonymized and appropriate consent forms should be signed.

*The study has been approved by a local ethics committee and participants were fully

debriefed after the study and given the possibility to withdraw their consent. Data have been sufficiently anonymized. However, as already outlined above, there might be provided some more information on the concrete open-ended response variables that have been anonymized as well as on the applied anonymization techniques (i.e., besides deletion of variables).